# Tumor-Tissue Expression of the Hyaluronic Acid Receptor RHAMM Predicts Histological Transformation in Follicular Lymphoma Patients

**DOI:** 10.3390/cancers14051316

**Published:** 2022-03-04

**Authors:** Marie Beck Enemark, Trine Engelbrecht Hybel, Charlotte Madsen, Kristina Lystlund Lauridsen, Bent Honoré, Trine Lindhardt Plesner, Stephen Hamilton-Dutoit, Francesco d’Amore, Maja Ludvigsen

**Affiliations:** 1Department of Hematology, Aarhus University Hospital, 8200 Aarhus, Denmark; mariem@rm.dk (M.B.E.); trihyb@rm.dk (T.E.H.); charmaen@rm.dk (C.M.); frandamo@rm.dk (F.d.); 2Department of Clinical Medicine, Aarhus University, 8000 Aarhus, Denmark; 3Department of Pathology, Aarhus University Hospital, 8000 Aarhus, Denmark; krislaur@rm.dk (K.L.L.); stephami@rm.dk (S.H.-D.); 4Department of Biomedicine, Aarhus University, 8000 Aarhus, Denmark; bh@biomed.au.dk; 5Department of Pathology, Copenhagen University Hospital, 2100 Copenhagen, Denmark; trine.lindhardt.plesner@regionh.dk

**Keywords:** follicular lymphoma (FL), histological transformation (HT), receptor for hyaluronan mediated motility (RHAMM), CD44

## Abstract

**Simple Summary:**

Histological transformation remains the leading cause of death in patients diagnosed with follicular lymphoma (FL). To date, no clinical nor biological biomarkers have been identified to unequivocally predict patients in high risk of transformation. In this study, we investigated the predictive value of the hyaluronic acid receptors RHAMM and CD44. Expression levels of RHAMM were higher in patients with subsequent transformation and were associated with poorer outcome.

**Abstract:**

Histological transformation (HT) remains the leading cause of mortality in follicular lymphoma (FL), underlining the need to identify reliable transformation predictors. The hyaluronic acid receptors CD44 and the receptor for hyaluronan mediated motility (RHAMM, also known as HMMR and CD168), have been shown to be involved in the pathogeneses of both solid tumors and hematological malignancies. In an attempt to improve risk stratification, expression of RHAMM and CD44 were evaluated by immunohistochemistry and digital image analysis in pre-therapeutic tumor-tissue biopsies from FL patients, either without (nt-FL, *n* = 34), or with (st-FL, *n* = 31) subsequent transformation, and in paired biopsies from the transformed lymphomas (tFL, *n* = 31). At the time of initial diagnosis, samples from st-FL patients had a higher expression of RHAMM compared with samples from nt-FL patients (*p* < 0.001). RHAMM expression further increased in tFL samples following transformation (*p* < 0.001). Evaluation of CD44 expression showed no differences in expression comparing nt-FL, st-FL, and tFL samples. Shorter transformation-free survival was associated with high tumoral and intrafollicular RHAMM expression, as well as with low intrafollicular CD44 expression (*p* = 0.002, *p* < 0.001, and *p* = 0.034, respectively). Our data suggest that high tumor-tissue RHAMM expression predicts the risk of shorter transformation-free survival in FL patients already at initial diagnosis.

## 1. Introduction

Follicular lymphoma (FL) is a lymphoproliferative neoplasia arising from germinal center B cells, comprising approximately 20% of all non-Hodgkin lymphomas. The disease is usually characterized by an indolent course with a median survival in excess of 10 years. However, FL is generally considered incurable because most patients eventually experience progression with recurrent relapses. In addition, a subset of patients experience early disease progression and treatment refractoriness [1,2,3,4,5]. Moreover, histological transformation (HT) to a more aggressive lymphoma, typically diffuse large B-cell lymphoma (DLBCL), remains the leading cause of FL-related mortality [3,6]. Although the cumulative incidence of HT has decreased after the introduction of CD20-targeting therapy, transformation markedly reduces the response to treatment and results in the reduction in median survival time after transformation of approximately 1–2 years [7,8,9,10,11]. No single biological event or mechanism has been shown to account for HT; thus, the observed clinical heterogeneity may reflect different underlying molecular mechanisms. To date, no clinical nor biological markers have been identified as unequivocal predictors of HT [11]. Given the markedly poorer clinical course associated with HT, identification of reliable predictors of transformation would be valuable.

It has been suggested that in the transformation event both intrinsic tumor cells and the tumor microenvironment (TME) play an important role [12,13]. The microenvironment is involved in controlling both inflammatory and malignant processes, and strong evidence indicates that the TME can regulate tumor growth [14]. In the present study, we aimed to examine expression levels of CD44 and receptor for hyaluronic acid mediated motility (RHAMM, also known as hyaluronan mediated motility receptor/HMMR and CD168) in relation to FL transformation. Both markers are receptors of the glycosaminoglycan hyaluronic acid (HA), which is a major component of extracellular matrices [14]. In previous studies, CD44 and RHAMM have been shown to act either on their own, together, or in the presence of HA to form trimeric complexes. These complexes have been proposed to play pivotal roles in cell proliferation, migration, and invasion, making them receptors of interest in the transformation of indolent to aggressive lymphomas [14,15].

Although CD44 is encoded by a single gene, alternative splicing generates multiple CD44 variant isoforms. The standard CD44 isoform (CD44s) is found in most cells, whereas the variant isoforms, with a variable number of exon insertions (designated v1–v10), are primarily expressed on cells during inflammation, lymphocyte maturation and activation, and in several types of tumor cells [14,16]. Studies have indicated that different CD44 variant isoforms play a crucial role in tumor progression. Their expression in tumors correlates with a poorer prognosis in cancer patients, including those with various hematological malignancies including some non-Hodgkin lymphomas, although this remains unexplored in FL with and without HT [17,18,19,20,21,22,23,24,25,26].

RHAMM has previously been found upregulated in several solid tumors, and higher expression of RHAMM has been associated with poorer clinical outcomes [27,28,29,30,31,32]. Under homeostatic conditions, RHAMM expression is generally very low, although its expression is increased during pathological conditions such as inflammation and cancer. This makes RHAMM an interesting target for low toxicity cancer therapy [14,33]. In hematological neoplasia, RHAMM is considered a possible target for tumor immunotherapy, as it has been identified as a tumor-associated antigen expressed in a broad variety of hematological malignancies [34,35,36,37,38]. Furthermore, clinical trials with RHAMM peptide vaccination have demonstrated clinical and immunological responses in patients with acute myeloid leukemia (AML), chronic lymphocytic leukemia (CLL), multiple myeloma (MM), and myelodysplastic syndrome (MDS) [39,40,41]. Co-expression of RHAMM and CD44 was found to be predictive of poorer outcomes in DLBCL, and RHAMM expression was shown to be a negative prognostic marker in pediatric leukemia [42,43]. Otherwise, RHAMM’s role in lymphoid neoplasia, including FL, has not yet been elucidated.

In the present study, CD44 and RHAMM expression levels were evaluated in pretherapeutic tumor tissues from FL patients, both with and without subsequent HT. Furthermore, paired tumor samples from the time of HT were included in the analyses to investigate possible changes in expression levels, from the time of initial FL diagnosis to the time of HT [7].

## 2. Materials and Methods

### 2.1. Patient Samples

Analyses were performed on pre-therapeutic formalin-fixed, paraffin embedded (FFPE) tumor tissue specimens from 65 FL patients diagnosed with a primary diagnosis of FL grade 1–3A at Aarhus University Hospital, Denmark, between 1990 and 2015. These included 34 patients who had no medical record of transformation for at least 10 years or until death (non-transforming FL, nt-FL) and 31 patients with subsequent histologically confirmed transformation to DLBCL or FL grade 3B, at least 6 months after the primary FL diagnosis (subsequentially-transforming FL, st-FL). In addition, for the 31 st-FL patients, paired transformed lymphoma samples from the time of HT were also analyzed (histologically transformed FL, tFL). All biopsies were reviewed by two experienced hematopathologists (SHD and TLP) and classified according to the 2017 update on the WHO Classification of Tumors of the Haematopoietic and Lymphoid Tissues [5]. Both hematopathologists agreed on all diagnoses and grades of the final cohort. Clinicopathological data on all patients were obtained from the Danish Lymphoma Registry (LYFO) [44] and, when relevant, patients medical records. Both clinicopathological and immunohistochemical data on other putative biological markers in this cohort have been published previously [7,10,45,46]. The study was approved by the Danish National Committee on Health Research Ethics (1-10-72-276-13) and the Danish Data Protection Agency (1-16-02-407-13) and was conducted in accordance with the Declaration of Helsinki.

### 2.2. Immunohistochemical Staining for RHAMM and CD44

Immunohistochemical staining for RHAMM and CD44 was performed on 4 μm FFPE sections using the Ventana Benchmark Ultra automated staining system (Ventana Medical Systems, Oro Valley, AZ, USA) using standard methods. Slides were deparaffinized with EZ Prep (Ventana, 950-102) followed by the blocking of endogenous peroxidase activity using a 3.0% hydrogen peroxide solution from the OptiView DAB IHC Detection Kit (Ventana, 760-700) [7,47]. For RHAMM staining, heat induced epitope retrieval (HIER) was applied by heating slides to 100 °C for 32 min in ULTRA Cell Conditioning Solution 1 (Ventana, 950-224). Primary polyclonal rabbit anti-human anti-RHAMM antibody (HPA040025, Sigma-Aldrich, St. Louis, MO, USA) was diluted 1:2000 in Tris buffered antibody diluent (pH 7.2, 15 mmol/L NaN3 and stabilizing protein, Dako, Santa Clara, CA, USA) followed by 32 min of incubation at 37 °C. For CD44 staining, HIER was applied by heating slides to 100 °C for 24 min. Primary monoclonal mouse anti-human antibody, recognizing standard form CD44s (Clone 156-3C11, Neomarkers, Thermo Fisher Scientific, Waltham, MA, USA), was diluted 1:400 in Tris buffered antibody diluent and incubated for 16 min at 37 °C. Visualization was performed using the OptiView DAB IHC Detection Kit (Ventana, 760-700) with nuclear counterstaining by hematoxylin. Sections of appendix, tonsil, liver, and pancreas were included on all slides as positive and negative controls [7,47].

### 2.3. Digital Image Analysis

Stained slides were scanned at a magnification of ×40 using a Hamamatsu Nanozoomer 2.0HT scanner (Hamamatsu, Shizouka, Japan), creating digital images of the stained tissue sections. Expression levels of RHAMM and CD44 were quantified using a Visiopharm Integrator system 2020.01 (Visiopharm A/S, Hoersholm, Denmark). In short, and as previously shown [7], areas of tissue suitable for staining quantification were defined by the manual outlining of regions of interest (ROI) on each digitized whole tissue section. Distinct areas of non-lymphoid tissue and technical artefacts were excluded. An analysis protocol package (APP) was designed to quantify the expression levels of each marker [7,46]. Staining quantification outputs were area fractions (AFs), defined as the stained area normalized to the total area within the ROI. For both RHAMM and CD44, expression levels were calculated on AFs of all positive staining [7,47]. Intrafollicular regions were manually outlined, guided by a consecutive parallel tissue section stained with PAX5 to identify B cell areas in the biopsy [7]. For the intrafollicular quantification, a total of 6 samples were excluded from the cohort due to the inability to define follicles based on the consecutive PAX5 staining (*n* = 59; nt-FL, *n* = 33 and st-FL, *n* = 26).

### 2.4. Statistical Analysis

Differences in AFs and CD44/RHAMM ratios of nt-FL, st-FL, and tFL samples were assessed using an independent Mann–Whitney U test and a paired Wilcoxon ranked sum test. Differences in clinicopathological features were assessed using a chi-squared test and Fisher’s exact test. Correlation of biomarker expression and clinicopathological features was evaluated using a Spearman’s rank test. Time-related endpoints were analyzed using the Kaplan–Meier and log rank method with overall survival (OS), progression-free survival (PFS), and transformation-free survival (TFS) as endpoints. OS was defined as time from initial FL diagnosis to the date of death by any cause or censoring. PFS was defined as time from initial FL diagnosis to the date of progression, relapse, HT, death, or censoring. TFS was defined as time from initial FL diagnosis to the date of biopsy-proven HT or censoring [7,10]. Cutoff values for high versus low expression of RHAMM and CD44, respectively, for OS, PFS, and TFS analyses were determined by a ROC analysis with the optimal cutoff point calculated using Youden’s index. The effect of potential confounders regarding OS and PFS on the cause-specific hazards were estimated in both a univariate and adjusted multivariate analysis using a Cox proportional hazards model. To account for missing data, multiple imputations were performed under the assumption that data were missing at random. *p*-values below 0.05 were considered statistically significant. Statistical analyses were performed using R Statistical Software (version 4.1.0).

## 3. Results

### 3.1. RHAMM but Not CD44 Expression Predicts HT in Follicular Lymphoma

The patient cohort comprised a total of 65 FL patients, including 34 nt-FL patients and 31 st-FL patients, Table 1. The study included 34 males and 31 females; the age at diagnosis ranged from 35 to 78 years with a median age of 55 years. Patients with subsequent transformation had higher risk profiles compared with nt-FL patients, with more advanced Ann Arbor stage and higher FLIPI score. In addition, significantly more st-FL patients had bone marrow involvement and elevated LDH levels compared with nt-FL patients.

Immunohistochemical evaluation of tumor-tissue RHAMM expression revealed cytoplasmatic/membranous staining of cellular subsets within the tumor samples, primarily located in follicular areas, Figure 1A,B. At the time of initial FL diagnosis, samples from st-FL patients had significantly higher expression of RHAMM compared with samples from nt-FL patients (*p* < 0.001), Figure 1C. Expression of RHAMM further increased at the time of HT, with significantly higher levels in tFL samples compared with st-FL samples (*p* < 0.001), Figure 1C. When quantifying RHAMM expression exclusively localized within intrafollicular areas, this significant difference was retained (*p* = 0.021), Figure 1D. We found a significant correlation to FLIPI scores (ρ = 0.27, *p* = 0.037) and a trend towards a correlation to the presence of bone marrow involvement (ρ = 0.22, *p* = 0.064), indicating a weak positive correlation between RHAMM levels and these clinicopathological parameters. RHAMM expression levels did not correlate with other included clinicopathological features. 

Immunohistochemical evaluation of CD44 showed diffuse intracellular staining of both neoplastic and non-neoplastic cells in the TME. Staining generally showed higher intensities in interfollicular areas, Figure 1E,F. Quantification of immunohistochemical staining showed no significant differential expression in nt-FL, st-FL, nor tFL samples, Figure 1G. Interestingly, while not differentially expressed, CD44 showed a wide range in expression in all patient groups, from almost no expression to very high levels, indicating high tumor heterogeneity. When analyzing exclusively in intrafollicular areas, no difference in expression was seen between patients with or without subsequent HT, Figure 1H. There was no correlation between CD44 expression and any of the clinicopathological features studied.

### 3.2. The Relationship between CD44 and RHAMM Expression Predicts HT in FL

To investigate the relationship between the two HA receptors, we calculated the ratio between the tumor-tissue levels of CD44 and RHAMM. The median CD44/RHAMM ratio in all nt-FL and st-FL tumors was 84.62 (range 0.10–4369.62), indicating that, in general, CD44 was much more abundantly expressed in the specimens. In samples from nt-FL patients, CD44/RHAMM ratios were significantly higher compared with samples from st-FL patients (medians 113.96 and 42.76, respectively, *p* = 0.008), Figure 1I. After transformation, tFL samples had significantly lower CD44/RHAMM ratios compared with st-FL samples (*p* = 0.003) with a median ratio of 9.46, Figure 1I. This reflects the finding of significantly higher RHAMM levels in st-FL and especially tFL samples compared with nt-FL samples.

When only evaluating intrafollicular expression, the median CD44/RHAMM ratio in nt-FL and st-FL tumors was 16.11 (range 0.01–1013.31), which is consistent with the observation of a generally higher RHAMM expression and lower CD44 expression within the follicles. CD44/RHAMM ratios were significantly higher in nt-FL compared with st-FL samples (medians 32.69 and 10.67, respectively, *p* = 0.024), Figure 1J.

### 3.3. High Tumor-Tissue Expression of RHAMM and Low Intrafollicular CD44 Expression Predicts Shorter Transformation-Free Survival

High levels of RHAMM expression at the time of initial FL diagnosis were found to be associated with a significantly shorter OS and TFS (cutoff AF = 0.0028, *p* = 0.037 and cutoff AF = 0.0051, *p* = 0.002, respectively) when analyzing the whole tumor section, Figure 2A,B. When exclusively analyzing intrafollicular RHAMM expression, OS, PFS, and TFS were all significantly shorter among patients with high marker levels (cutoff AF = 0.0096, *p* = 0.032; cutoff AF = 0.0206, *p* = 0.028; and cutoff AF = 0.0096, *p* < 0.001, respectively), Figure 2C–E.

High versus low expression levels of CD44 in whole tumor sections at initial FL diagnosis were not associated with patient OS, PFS, nor TFS when comparing patients with high versus low expression. When exclusively analyzing intrafollicular CD44 expression, TFS was significantly shorter among patients with lower CD44 levels (cutoff AF = 0.0034, *p* = 0.011), while differences in OS and PFS were only trending (cutoff AF = 0.1572, *p* = 0.184 and cutoff AF = 0.3755, *p* = 0.074, respectively), Figure 2F–H. However, for the intrafollicular TFS analysis, only a few patients were in the low expression category, thereby reducing the statistical power.

Lower CD44/RHAMM ratios in whole tumor sections were found to be associated with a shorter TFS (cutoff AF = 125.76, *p* = 0.048), Figure 2I. This was also seen with CD44/RHAMM ratios in intrafollicular areas (cutoff AF = 34.52, *p* = 0.034), Figure 2J–K. This observation supports our previous findings, indicating that primarily RHAMM expression levels are correlated with a poorer prognosis in FL patients.

With a median follow up time of 15.6 years (range 0.9–24.5), univariate and multivariate analyses were performed regarding OS and PFS, Table 2. In the univariate analysis, the cause-specific hazard for OS was significantly elevated in patients with higher Ann Arbor stage, high FLIPI risk score, and LDH elevation. In the multivariate analysis, only a trend towards LDH elevation was seen. Regarding PFS, the univariate analysis, the hazard was significantly elevated in patients with higher Ann Arbor stage, high FLIPI score, LDH elevation, and bone marrow involvement. In the multivariate analysis, higher Ann Arbor stage and LDH elevation retained this significance, Table 2.

## 4. Discussion

We show that expression levels of the two HA receptors, RHAMM and CD44, in diagnostic FL tumor-tissue samples from patients with and without subsequent transformation to a high-grade lymphoma can predict HT.

In particular, we found that RHAMM expression levels were increased at primary FL diagnosis in tumor-tissue from patients who subsequently experienced HT. At HT diagnosis, RHAMM expression levels were further increased in tFL samples, and may thus reflect the increasing aggressiveness of the tumors, the hypothesis being that st-FL tumors show a more aggressive phenotype at FL diagnosis than nt-FL.

To the best of our knowledge, this is the first study that has identified RHAMM expression as a possible predictive biomarker of transformation in FL. This finding makes the marker an interesting candidate with regard to FL treatment, because RHAMM has already been identified as a suitable target for cancer therapy, with low toxicity shown in clinical trials of RHAMM vaccination in other hematological malignancies [14,39,40,41]. In these trials, AML, MDS, MM, and CLL patients were vaccinated with the RHAMM-derived peptide R3, which is a highly immunogenic CD8+ T cell epitope. Positive immunological and clinical responses were seen with a specific lysis of RHAMM+ leukemic blasts, a reduction in the number of leukemic blasts in the bone marrow, and increased numbers of RHAMM-R3-specific CD8+ T cells. The studies reported no drug-induced adverse events higher than CTC grade 1 skin toxicity [39,40,41]. In the light of these results, and our findings in the present study, RHAMM constitutes a promising target for immunotherapy in lymphoid malignancies, and further investigations on a possible functional role of RHAMM in the process of histological transformation in FL are warranted.

We found no difference in expression of CD44, which showed wide variability in levels, indicating substantial tumor heterogeneity. Nonetheless, despite the heterogeneous expression found in the samples, we did see a correlation with prognosis and survival, in part, based on CD44 alone, but more strongly based on combined expression levels with RHAMM. At FL diagnosis, high levels of RHAMM and low expression of CD44 correlated with adverse prognosis and survival. In cancer in general, CD44 is regarded as a tumor promoting protein, acting through, amongst others, interactions with HA and the PI3K-AKT pathway [48]. HA holds a crucial role in cancer cell survival, proliferation, and invasion, which can be induced through interactions with both RHAMM and CD44, acting through PI3K, MAPK, NFκB, and RAS, as well as cytoskeletal components required for cancer development. However, the potential mechanisms of interaction between RHAMM and CD44 are not clearly understood [14]. Our results could indicate a dominating role of RHAMM over CD44 in the case of transformation of FL.

As described, various isoforms of CD44 exist [14,16], and the antibody used for immunohistochemical analysis in this study is expected to recognize all isoforms. Given that specific CD44 isoforms have previously been associated with tumor promoting properties [48], future studies might be aimed at elucidating the possible role of individual isoforms with regard to HT in FL.

Importantly, our study is based on analyses performed on tumor-tissue sections, with defined ROIs delineating lymphoid tissue, thus evaluating both neoplastic and non-neoplastic cells of the TME. The composition of the TME influences outcome in patients with FL [12], and the underlying FL tumor biology as a whole is believed to represent a phenotype that supports the transformation process. Immunohistochemical staining of both RHAMM and CD44 showed differences in expression levels in both inter- and intrafollicular areas. In FL, the tissue architecture is different when comparing inter- and intrafollicular areas, with regard to not only the presence and numbers of tumor cells, but also to the makeup of the TME.

For this study, the samples size was limited to patients with sufficient FFPE tumor biopsies available. Inclusion based on availability introduced differences in risk profiles between tumors from nt-FL and st-FL patients with regard to Ann Arbor stage, FLIPI score, LDH elevation, and bone marrow involvement. These differences could potentially bias analyses, and, therefore, increased expression levels of RHAMM resulting from, for example, stage differences cannot be ruled out. RHAMM expression levels did show a correlation to FLIPI scores, but otherwise, RHAMM and CD44 expression levels were not correlated with any clinicopathological features. In tFL samples, RHAMM expression levels were elevated at transformation to a high-grade lymphoma as compared with st-FL samples. This may indicate RHAMM as a marker of aggressive disease, given that the high-grade lymphomas would be expected to have a more adverse prognosis compared with FL. Furthermore, we found no association between treatment of FL and subsequent HT, which may be due to the smaller sample size. In order to fully investigate this question, larger cohort studies including clinicopathological data should be performed.

Targeting tumor cell surface CD20 with rituximab has markedly demonstrated its benefits in the prognosis of FL patients; however, the management of FL and transformed FL continues to pose a clinical challenge [49]. With the increasing understanding of the FL tumor biology, development of a range of novel therapies targeting the tumor is warranted. Our results suggest a new and interesting role for the two HA receptors, CD44 and RHAMM, as biomarkers of progression and transformation of FL. The exact interaction between the two receptors, and their function in the transformation process, should be investigated further and validated in larger cohorts.

## 5. Conclusions

Our study of the HA receptors RHAMM and CD44 in the transformation of FL is the first to identify RHAMM as a possible predictive biomarker of HT in FL patients. Expression of both biomarkers was assessed at time of FL diagnosis by standard immunohistochemical methods, which are widely available in routine diagnostic pathology laboratories and thus are easily implementable. The study further suggests a correlation of both markers with prognosis and survival in FL patients. Our findings warrant validation in larger and independent cohorts, which could support the prognostic and/or predictive roles of CD44 and RHAMM, making them useful future tools for early risk-adapted management of FL patients.

## Figures and Tables

**Figure 1 cancers-14-01316-f001:**
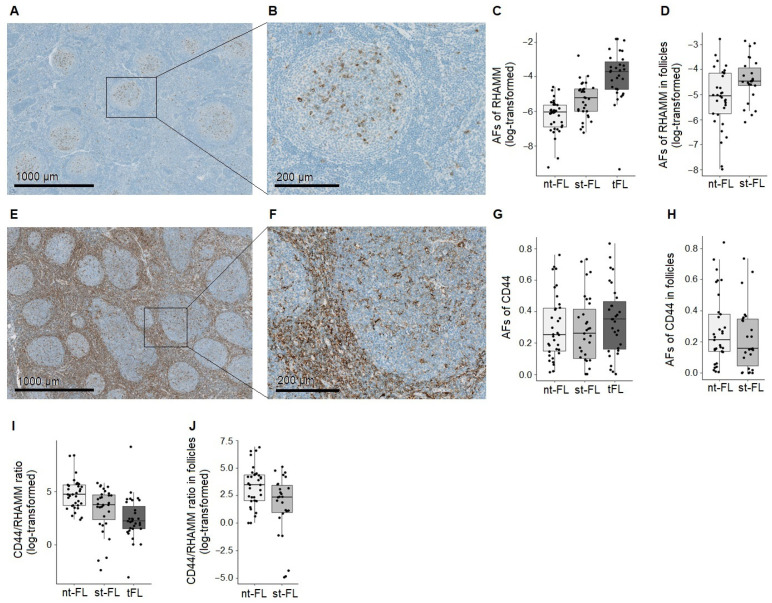
RHAMM and CD44 in follicular lymphoma. (**A**,**B**) Representative images of tumor tissue stained for RHAMM. (**C**) Area fractions of RHAMM staining in FL diagnostic samples from patients with and without subsequent HT, and in lymphoma samples from time of HT diagnosis. (**D**) Area fractions of RHAMM staining exclusively localized in intrafollicular areas of diagnostic FL samples from patients with and without subsequent HT. (**E**,**F**) Representative images of tumor tissue stained for CD44. (**G**) Area fractions of CD44 staining in FL diagnostic samples from patients with and without subsequent HT, and in lymphoma samples from time of HT diagnosis. (**H**) Area fractions of CD44 staining exclusively localized in intrafollicular areas of diagnostic FL samples from patients with and without subsequent HT. (**I**) CD44/RHAMM ratio calculated from expression of both markers in the tumor-tissue sections for patients with and without subsequent HT, and in lymphoma samples from time of HT diagnosis. (**J**) CD44/RHAMM ratio calculated from intrafollicular expression of both markers for patients with and without subsequent HT. Abbreviations: nt-FL, non-transforming follicular lymphoma; RHAMM, receptor of hyaluronic acid mediated motility; st-FL, subsequently-transforming follicular lymphoma; tFL, histologically transformed follicular lymphoma.

**Figure 2 cancers-14-01316-f002:**
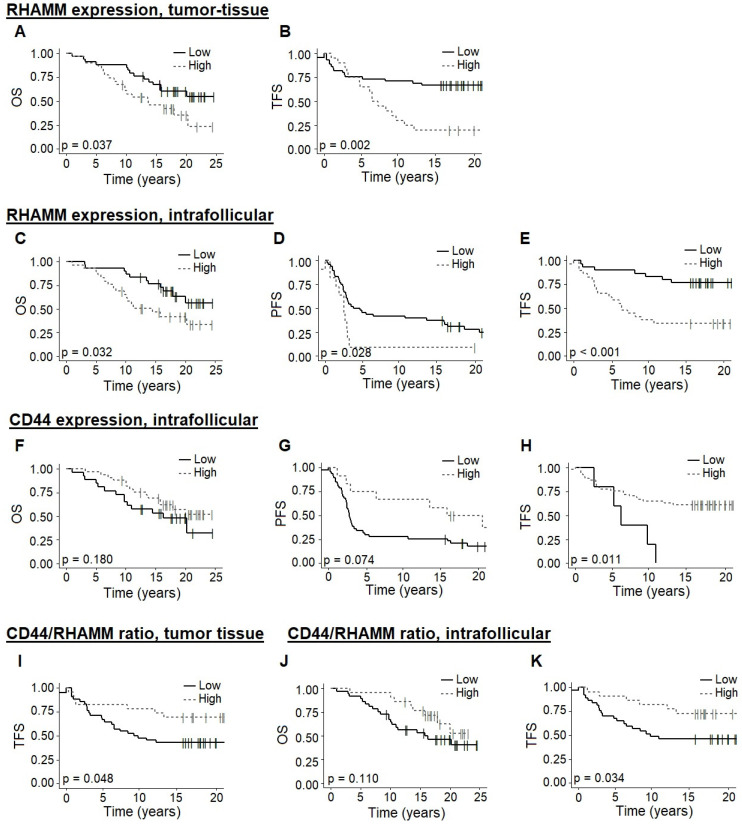
Outcome according to RHAMM and CD44. (**A**,**B**) Association between RHAMM expression in tumor-tissue biopsies and OS (cutoff AF = 0.0028) and TFS (cutoff AF = 0.0051), respectively. (**C**–**E**) Association between intrafollicular RHAMM expression and OS (cutoff AF = 0.0096), PFS (cutoff AF = 0.0206), and TFS (cutoff AF = 0.0096), respectively. (**F**–**H**) Association between exclusively intrafollicular CD44 expression and OS (cutoff AF = 0.1572), PFS (cutoff AF = 0.3755), and TFS (cutoff AF = 0.0034), respectively. (**I**) Association between CD44/RHAMM ratio in tumor-tissue biopsies and TFS (cutoff AF = 125.76). (**J**,**K**) Association between intrafollicular CD44/RHAMM ratio and OS (cutoff = 34.52) and TFS (cutoff AF = 34.52), respectively. Abbreviations: OS, overall survival; PFS, progression-free survival; TFS, transformation-free survival.

**Table 1 cancers-14-01316-t001:** Patients’ clinicopathological features.

Characteristics	All *n* = 65 *n* (%)	nt-FL *n* = 34 *n* (%)	st-FL *n* = 31 *n* (%)	*p*-Value
Sex				NS
Male	34 (52)	16 (47)	18 (42)	
Female	31 (48)	18 (53)	13 (58)	
Age at FL diagnosis				NS
Median	55	54	57	
Range	35–78	35–76	40–78	
Ann Arbor stage				<0.001
I–II	17 (26)	15 (44)	2 (6)	
III–IV	46 (71)	18 (53)	28 (90)	
Unknown	2 (3)	1 (3)	1 (3)	
FLIPI				<0.001
Low	24 (37)	20 (59)	4 (13)	
Intermediate	18 (28)	10 (29)	8 (26)	
High	19 (29)	2 (6)	17 (55)	
Unknown	4 (6)	2 (6)	2 (6)	
LDH-elevation				0.022
Yes	8 (12)	1 (3)	7 (23)	
No	53 (82)	31 (91)	22 (71)	
Unknown	4 (6)	2 (6)	2 (6)	
B-symptoms				NS
Yes	15 (23)	6 (18)	9 (29)	
No	47 (72)	27 (79)	20 (65)	
Unknown	4 (6)	1 (3)	2 (6)	
Performance score				NS
<2	49 (75)	28 (82)	21 (68)	
≥2	13 (20)	5 (15)	8 (26)	
Unknown	3 (5)	1 (3)	2 (6)	
Bone marrow Involvement				0.018
Yes	20 (31)	6 (18)	14 (45)	
No	36 (55)	24 (71)	12 (39)	
Unknown	9 (14)	4 (12)	5 (16)	
Anemia				NS
Yes	5 (8)	1 (3)	4 (13)	
No	57 (88)	32 (94)	25 (81)	
Unknown	3 (5)	1 (3)	2 (6)	
FL histology				NS
FL grade 1–2	56 (86)	29 (85)	27 (87)	
FL grade 3A	9 (14)	5 (15)	4 (13)	
RHAMM expression, biopsy				0.019
Low	34 (52)	23 (68)	11 (35)	
High	31 (48)	11 (32)	20 (65)	
CD44 expression, biopsy				NS
Low	26 (40)	14 (41)	12 (39)	
High	39 (60)	20 (59)	19 (61)	
Ratio CD44/RHAMM expression, biopsy				0.072
Low	42 (65)	18 (53)	24 (77)	
High	23 (35)	16 (47)	7 (23)	
RHAMM expression, intrafollicular *				NS
Low	42 (71)	23 (70)	19 (73)	
High	17 (29)	10 (30)	7 (27)	
CD44 expression, intrafollicular *				NS
Low	26 (44)	12 (36)	14 (54)	
High	33 (56)	21 (67)	12 (46)	
Ratio CD44/RHAMM expression, intrafollicular *				0.083
Low	37 (63)	17 (52)	20 (77)	
High	22 (37)	16 (48)	6 (23)	
Initial treatment				NA
Alkylator-based	22 (34)	8 (24)	14 (45)	
Antracyclin-based	21 (32)	12 (35)	9 (29)	
Rituximab	19 (29)	6 (18)	13 (42)	
Observation only	4 (6)	3 (9)	1 (3)	
R-Chemotherapy	9 (14)	1 (3)	8 (28)	
Other	5 (8)	2 (6)	3 (10)	
Unknown	19 (29)	11 (32)	8 (26)	

Analyses of RHAMM and CD44 on tumor-tissue biopsies were performed with the OS cutoff for high versus low expression for both markers based on ROC analyses (cutoffs at AF = 0.0028 and AF = 0.2012, respectively). Analyses of intrafollicular RHAMM and CD44 expression were performed with the OS cutoff (AF = 0.0096 and AF = 0.1572, respectively). * For the intrafollicular analyses, a total of 6 samples were excluded from the study. The cohort reduction did not affect the cohort characteristics. Analyses of CD44/RHAMM ratios were performed with the TFS cutoff. Clinicopathological data for this cohort has been published previously [7,10,45]. Abbreviations: FLIPI, follicular lymphoma international prognostic index; LDH, lactate dehydrogenase; NA, not applicable; NS, not significant; nt-FL, non-transforming follicular lymphoma; RHAMM, receptor of hyaluronic acid mediated motility; st-FL, subsequently-transforming follicular lymphoma.

**Table 2 cancers-14-01316-t002:** Univariate and multivariate analysis for OS and PFS.

Clinicopathological Feature	Univariate *p* and HR (95% CI) Values				Multivariate *p* and HR (95% CI) Values			
	OS		PFS		OS		PFS	
Sex, male	NS	0.9 (0.5–1.8)	NS	1.0 (0.6–1.7)	NS	1.3 (0.6–3.0)	NS	1.3 (0.7–2.4)
Age above 60	NS	1.7 (0.8–3.3)	NS	1.7 (0.9–3.0)	NS	1.6 (0.6–3.9)	NS	1.3 (0.6–2.7)
Ann Arbor stage III–IV	0.016	1.6 (1.1–2.2)	0.005	1.4 (1.1–1.8)	NS	1.5 (0.9–2.5)	0.021	1.5 (1.1–2.1)
High FLIPI risk score	0.009	2.9 (1.3–6.3)	0.006	2.6 (1.3–5.3)	NS	1.8 (0.7–4.5)	0.058	2.3 (0.9–5.5)
Elevated LDH	0.033	2.9 (1.1–7.6)	0.004	3.4 (1.5–7.7)	0.056	3.2 (0.9–10.3)	0.006	3.9 (1.5–10.5)
B-symptoms	NS	1.5 (0.7–3.3)	NS	1.5 (0.8–2.8)	NS	1.1 (0.4–2.8)	NS	1.0 (0.5–2.2)
Performance score ≥ 2	NS	2.0 (0.9–4.3)	NS	1.4 (0.7–2.6)	NS	2.0 (0.8–5.3)	NS	1.6 (0.7–3.4)
Anemia	NS	1.6 (0.5–5.3)	NS	1.5 (0.5–4.3)	NS	0.7 (0.2–2.7)	NS	0.7 (0.2–2.2)
Bone marrow involvement	NS	1.9 (0.9–4.2)	0.043	1.9 (1.0–3.4)	NS	1.0 (0.4–2.6)	NS	0.9 (0.4–1.9)
FL grade 3A	NS	0.9 (0.3–2.7)	NS	1.1 (0.5–2.6)	NS	1.3 (0.4–4.7)	NS	1.4 (0.5–4.1)

## Data Availability

These data analyzed during the current study are available upon reasonable request.

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
