# Peer review of "Tumor-Tissue Expression of the Hyaluronic Acid Receptor RHAMM Predicts Histological Transformation in Follicular Lymphoma Patients"

_cancers, 2022, doi:10.3390/cancers14051316_

Round 1

Reviewer 1 Report

Enenark et al. tackle an important issue in follicular lymphoma, i.e. the development of biomarkers relevant for and predictive of histologic transformation to aggressive lymphoma. In particular, the authors longitudinally investigate the expression of the hyaluronic acid receptors RHAMM and CD44 in paired samples of follicular lymphoma and transformed follicular lymphoma. The data document that expression of RHAMM at the time of follicular lymphoma diagnosis predicts for subsequent development of histologic transformation.

MAJOR ISSUES

1) the authors state that all biopsies were reviewed by two experienced hematopathologists. What was the concordance rate of their reports?

2) did the authors detect any association between treatment for follicular lymphoma and subsequent transformation?

3) was the level of RHAMM expression similar before and after transformation, or did any significant variation occur in the expression pattern/level?

4) In the Introduction, the authors should add a sentence on the current outcome of transformed follicular lymphoma and must mention the many novel drugs that are available or are in development (for a recent review on the topic, see: Patriarca A. et al, Investigational drugs for the treatment of diffuse large B-cell lymphoma. Expert Opin Investig Drugs. 2021:25-38)

MINOR ISSUES

1) line 182: I suppose that "7,10,45" refer to references. In that case, they should be formatted as all other citations in the text.

Author Response

MAJOR ISSUES

1) the authors state that all biopsies were reviewed by two experienced hematopathologists. What was the concordance rate of their reports?

Response: The two hematopathologists reviewed all biopsies independently of each other. If any of the hematopathologists disagreed on either FL, FL 3B, or transformed FL, these cases were excluded from the cohort. Thus, both hematopathologists agreed on all diagnoses and grades of the final cohort. This statement has been added to the “Patient samples” section (ll. 108-109).

2) did the authors detect any association between treatment for follicular lymphoma and subsequent transformation?

Response: We did investigate this; however, we found no difference, which may be due to the smaller number of included patients in the present study. In order to fully investigate this question, larger cohort studies including clinicopathological data should be performed. However, with the present study, we aimed to describe RHAMM and CD44 tumor tissue expression as the matter of interest.

This is indeed a relevant comment, and so, we have added a statement regarding this to the discussion (ll. 346-348).

3) was the level of RHAMM expression similar before and after transformation, or did any significant variation occur in the expression pattern/level?

Response: As mentioned in the results section (ll. 196-198) and shown in Figure 1C, RHAMM expression was significantly increased after the time of transformation (p<0.001).

4) In the Introduction, the authors should add a sentence on the current outcome of transformed follicular lymphoma and must mention the many novel drugs that are available or are in development (for a recent review on the topic, see: Patriarca A. et al, Investigational drugs for the treatment of diffuse large B-cell lymphoma. Expert Opin Investig Drugs. 2021:25-38)

Response: To highlight the markedly inferior outcome associated with transformation of follicular lymphoma, we have re-written the following sentence in the introduction.

“Although the cumulative incidence of HT has decreased after introduction of CD20-targeting therapy, transformation markedly reduces the response to treatment and results in the reduction in median survival time after transformation of approximately 1-2 years” (lines 46-49)

Here we also mention the anti-CD20 antibody, rituximab, which has currently shown great efficacy in treatment of both FL and transformed FL. Many novel drugs, which are in development, are indeed very interesting and important factors to discus, however, we believe that stating all of these would be rather noisy in the context of the present study, and we would like to maintain the focus of the study regarding the potential of RHAMM and CD44 as biomarkers in FL.

MINOR ISSUES

1) line 182: I suppose that "7,10,45" refer to references. In that case, they should be formatted as all other citations in the text.

Response: Thank you for noticing, this has been corrected.

Reviewer 2 Report

In Figure 1, the authors need to show both figures of RHAMM for FL with subsequent transformation and FL without subsequent transformation.

Author Response

In Figure 1, the authors need to show both figures of RHAMM for FL with subsequent transformation and FL without subsequent transformation.

Response: Figure 1A and 1B show a representative example of RHAMM staining across the FL biopsies. As seen in Figure 1C, quantified RHAMM expression for FL with and without subsequent transformation is presented. FL without subsequent transformation is designated “nt-FL”, FL with subsequent transformation is designated “st-FL”, and biopsies from time of HT is designated “tFL”.

Reviewer 3 Report

Marie Beck Enemark et al uncovered tumor-tissue expression of the hyaluronic acid receptor RHAMM to be potentially relevant in predicting histological transformation in FL.

Point to be considered:

  1. The authors show immunohistochemistry intensity (fig. 1). How did they quantify the expression (IMAGE J and similar, other software, pathology subjective evaluation?)
  2. In the frame of this thinking, how did the authors select the class boundaries between high and low?
  3. One major concern remains with regard to the separation of RHAMM-A high/low patient samples. The authors are kindly request to ass a more comprehensive explanation based on which patients were dichotomized into two classes, RHAMM high/low, choosing the median as class boundary and or quartiles. An additional figure (Supplementary ?) can be helpful, in order to confirm or rather weaken a dichotomized separation. It would be crucial to see the respective OS and PFS data for the lowest and highest quartile while sparing the two intermediate, mostly levelled quartiles. Adding this comparison will be very helpful to provide robust conclusions on RHAMM positive vs. negative in the clinical setting.
  4. Please add exact definition of RHAMM high/low in “Statistical Analysis” section of “Methods”. Please include a figure visualizing absolute RHAMM expression across the entire cohort.
  5. A multivariate analysis of possible confounding factors for PFS and OS should be performed. The authors should also clarify what was the median follow up of the patients included in the study.
  6. Finally, the authors state that "Although the cumulative incidence of HT has decreased after introduction of CD20-targeting therapy, transformation markedly reduces the response to treatment and survival time"and referenced n.7-10. This reviewer personally misses some insights regarding cell surface antigens overview that have been defined, that may be targets for therapy with monoclonal antibodies and radioimmunotherapy and that can boost the interest for the authors data fro a broad readership in the oncology field (please refer to PMID: 26818572)

Author Response

The authors show immunohistochemistry intensity (fig. 1). How did they quantify the expression (IMAGE J and similar, other software, pathology subjective evaluation?)

Response: As mentioned in the methods section, all stained slides were scanned using the Hamamatsu Nanozoomer 2.0HT scanner to create digital images of the stained tumor sections. Then protein expression of both RHAMM and CD44 were quantified by digital image analysis, to provide an objective staining quantification, as described in the “Digital image analysis” section.

In the frame of this thinking, how did the authors select the class boundaries between high and low?

Response: As mentioned in “Statistical analysis”, cutoff values for high versus low expression were determined by a ROC analysis, from which the optimal cutoff were calculated using Youden’s index. This was done separately for overall survival, progression-free survival, and transformation-free survival, respectively, in order to obtain the most statistically powerful separation for each analysis. The corresponding cutoff values are all listed in the figure legend of Figure 2. We are aware that these boundaries are sometimes chosen as the lower quartile, median, or upper quartile. However, when determined by a ROC analysis, we ensure using the most statistically optimal separation possible.

One major concern remains with regard to the separation of RHAMM-A high/low patient samples. The authors are kindly request to ass a more comprehensive explanation based on which patients were dichotomized into two classes, RHAMM high/low, choosing the median as class boundary and or quartiles. An additional figure (Supplementary ?) can be helpful, in order to confirm or rather weaken a dichotomized separation. It would be crucial to see the respective OS and PFS data for the lowest and highest quartile while sparing the two intermediate, mostly levelled quartiles. Adding this comparison will be very helpful to provide robust conclusions on RHAMM positive vs. negative in the clinical setting.

Response: As explained above, we are aware that the boundaries are sometimes based on the median or quartiles. However, to assure the optimal statistical separation, we calculated the cutoff values for both RHAMM and CD44 from a ROC analysis and Youden’s index. This was determined based on all patients, regardless of subsequent transformation the ensure the separation based on the biomarker expression and not based on the two defined patient groups. We have chosen to dichotomize patients into two groups in the manuscript in that the number of cases in each arm (if separated according to lower quartile, median, and upper quartile) are approaching the lower limit of sound statistical analyses.

Please add exact definition of RHAMM high/low in “Statistical Analysis” section of “Methods”.

Response: How high and low RHAMM (and CD44) were determined is already described in the statistical analysis section. The corresponding calculated cutoff values are listed in the figure legend of Figure 2. To clarify this, we have included the cutoff values in the results section along with the p-values as well as kept them in the figure legend.

Please include a figure visualizing absolute RHAMM expression across the entire cohort.

Response: We are not entirely sure which visualization the reviewer prefers. We do show absolute RHAMM expression on all biopsies included in Figure 1C, with a representative of staining pattern in Figures 1A-B.

A multivariate analysis of possible confounding factors for PFS and OS should be performed. The authors should also clarify what was the median follow up of the patients included in the study.

Response: With a median follow up time of 15.6 years (range 0.9-24.5), univariate and multivariate analyses were performed regarding OS and PFS. This has been added to the results section (lines 265-272) and are presented in the added Table 2.

In addition to this, the following statement was added to the “Statistical analysis” section: “The effect of potential confounders regarding OS and PFS on the cause-specific hazards were estimated in both a univariate and adjusted multivariate analysis using a Cox proportional hazards model. To account for missing data, multiple imputations were performed under the assumption that data were missing at random” (lines 164-168).

Finally, the authors state that "Although the cumulative incidence of HT has decreased after introduction of CD20-targeting therapy, transformation markedly reduces the response to treatment and survival time"and referenced n.7-10. This reviewer personally misses some insights regarding cell surface antigens overview that have been defined, that may be targets for therapy with monoclonal antibodies and radioimmunotherapy and that can boost the interest for the authors data fro a broad readership in the oncology field (please refer to PMID: 26818572)

Response: Thank you for suggesting this. We have added the following statement to the discussion section:

“Targeting tumor cell surface CD20 with rituximab has markedly demonstrated its benefits in the prognosis of FL patients, however, the management of FL and transformed FL continues to pose a clinical challenge. With the increasing understanding of the FL tumor biology, development of a range of novel therapies targeting the tumor is warranted.” (lines 349-353).

Round 2

Reviewer 1 Report

The authors have addressed the comments that had been made.

Reviewer 3 Report

The authors have clarified several of the questions I raised in my previous review. Most of the major problems have been addressed by this revision.